

# The complete mitochondrial genomes of five longicorn beetles (Coleoptera: Cerambycidae) and phylogenetic relationships within Cerambycidae

Jun Wang[1], Xin-Yi Dai[1], Xiao-Dong Xu[1], Zi-Yi Zhang[1], Dan-Na Yu[1,2], Kenneth B. Storey[3] and Jia-Yong Zhang[1,2]

[1] College of Chemistry and Life Science, Zhejiang Normal University, Jinhua, Zhejiang, China
[2] Key lab of wildlife biotechnology, Conservation and Utilization of Zhejiang Province, Zhejiang Normal University, Jinhua, Zhejiang, China
[3] Department of Biology, Carleton University, Ottawa, Ontario, Canada

## ABSTRACT

Cerambycidae is one of the most diversified groups within Coleoptera and includes nearly 35,000 known species. The relationships at the subfamily level within Cerambycidae have not been convincingly demonstrated and the gene rearrangement of mitochondrial genomes in Cerambycidae remains unclear due to the low numbers of sequenced mitogenomes. In the present study, we determined five complete mitogenomes of Cerambycidae and investigated the phylogenetic relationship among the subfamilies of Cerambycidae based on mitogenomes. The mitogenomic arrangement of all five species was identical to the ancestral Cerambycidae type without gene rearrangement. Remarkably, however, two large intergenic spacers were detected in the mitogenome of *Pterolophia* sp. ZJY-2019. The origins of these intergenic spacers could be explained by the slipped-strand mispairing and duplication/random loss models. A conserved motif was found between *trnS2* and *nad1* gene, which was proposed to be a binding site of a transcription termination peptide. Also, tandem repeat units were identified in the A + T-rich region of all five mitogenomes. The monophyly of Lamiinae and Prioninae was strongly supported by both MrBayes and RAxML analyses based on nucleotide datasets, whereas the Cerambycinae and Lepturinae were recovered as non-monophyletic.

## INTRODUCTION

Coleoptera (Hexapoda: Insecta) are a highly diverse group of insects consisting of about 360,000 known species of beetles that account for almost 40% of all described insect species (*Lawrence & Newton, 1982*; *Hunt et al., 2007*). Cerambycidae (longicorn beetles) is one of the species-rich families of Coleoptera and is a group of phytophagous insects with over 4,000 genera and 35,000 species in the world (*Monné, Monné & Mermudes, 2009*; *Sama et al., 2010*). Longicorn beetles are morphologically and ecologically diverse, and have significant effects on almost all terrestrial ecosystems (*Ponomarenko & Prokin, 2015*). Nevertheless, owing to their remarkable species richness, variable morphological features

Corresponding author
Jia-Yong Zhang,
zhangjiayong@zjnu.cn

and sparse gene data, the resolution of the phylogeny of longicorn beetles has turned out to be a difficult challenge (*Bologna et al., 2008*; *Zhang et al., 2018a*; *Zhang et al., 2018b*; *Zhang et al., 2018c*). Cerambycidae *s. s.* (sensu stricto) has usually been divided into eight subfamilies: Lamiinae, Cerambycinae, Lepturinae, Prioninae, Dorcasominae, Parandrinae, Spondylidinae and Necydalinae (*Svacha, Wang & Chen, 1997*) whereas Cerambycidae *s. l.* (sensu lato) was considered to consist of Cerambycidae *s. s.*, Disteniidae, Oxypeltidae and Vesperidae (*Napp, 1994*; *Reid, 1995*; *Svacha, Wang & Chen, 1997*). Even if the number and definition of Cerambycidae gradually stabilizes, the relationships at the subfamily level remained unclear.

The mitochondrial genome is widely considered to be an informative molecular marker for species identification, molecular evolution, and comparative genomic research (*Moritz, Dowling & Brown, 1987*; *Boore, 1999*) due to its maternal inheritance and high evolutionary rate properties (*Avise et al., 1987*). In the last few years, studies of animal mitogenomes have grown rapidly in number and approximately 40,000 mitogenome sequences have now been published in the NCBI database (*Tan et al., 2017*). By contrast, a mere 18 sequenced mitogenomes of Cerambycidae have been reported, among them being eight mitogenomes belonging to the subfamily Lamiinae, four mitogenomes of the subfamily Cerambycinae, three mitogenomes of the subfamily Prioninae, and three mitogenomes of the subfamily Lepturinae (*Kim et al., 2009*; *Chiu et al., 2016*; *Fang et al., 2016*; *Guo et al., 2016*; *Li et al., 2016a*; *Li et al., 2016b*; *Wang et al., 2016*; *Lim et al., 2017*; *Liu et al., 2017*; *Song et al., 2017*; *Liu et al., 2018*; *Que et al., 2019*; *Wang et al., 2019*). These few mitogenomes seriously restrict the capacity for phylogenetic analyses and phylogeography of the Cerambycidae.

The gene organization of the known mitogenomes of Coleoptera, especially the arrangements of protein-coding genes, are mostly in accordance with those of ancestral insects (*Timmermans & Vogler, 2012*). Nevertheless, recent evidence suggested that gene rearrangements had occurred in the tRNA of *Mordella atrata* (Coleoptera: Mordellidae) and *Naupactus xanthographus* (Coleoptera: Curculionidae) (*Song et al., 2010*). In addition to these, recombination in the control region was observed in *Phrixothrix hirtus* (Coleoptera: Phengodidae) and *Teslasena femoralis* (Coleoptera: Elateridae) (*Amaral et al., 2016*). The mitogenome structure was originally found with no introns, sparse intergenic spacers and no overlapping genes (*Ojala, Montoya & Attardi, 1981*). Nevertheless, large non-coding regions (except the A + T-rich region) in mitogenomes have been observed within beetles, including a 1724-bp long intergenic spacer region in *Pyrocoelia rufa* (Coleoptera: Lampyridae), a 494-bp region in *Hycleus chodschenticus* (Coleoptera: Meloidae) and two large intergenic spacers of more than 30 bp in *Hycleus* species (*Bae et al., 2004*; *Yuan et al., 2016*; *Haddad et al., 2018*). Previously reported tandem repeat units or an additional origin of replication were identified among large intergenic regions (*Dotson & Beard, 2001*; *Rodovalho et al., 2014*).

The phylogenetic relationships within Cerambycidae have yet to be fully resolved due to a lack of adequately convincing taxon sampling, and the monophyly of subfamilies within Cerambycidae need further discussion (*Haddad et al., 2018*; *Kim et al., 2018*). With the aim to discuss the monophyly of subfamilies of Cerambycidae and gene arrangements of the mitogenome, complete mitogenomes of the five longicorn beetle species were determined.

We also described the structural and compositional features of the newly sequenced mitogenomes and analyzed the intergenic spacers to explain the possible evolutionary mechanisms.

## MATERIALS AND METHODS

### Sampling collection and DNA extraction

Five longicorn beetle specimens (*Oberea yaoshana*, *Thermistis croccocincta*, *Blepephaeus succinctor*, *Nortia carinicollis*, *Pterolophia* sp. ZJY-2019) were captured from Jinxiu, Guangxi Zhuang Autonomous Region, China and were stored at −40 °C in the lab of JY Zhang (College of Chemistry and Life Science, Zhejiang Normal University). The specimens were identified by Dr. JY Zhang based on morphology. Total genomic DNA was extracted from the thorax muscle using Ezup Column Animal Genomic DNA Purification Kit (Sangon Biotech Company, Shanghai, China).

### PCR amplification and sequencing

In order to obtain the entire mitogenome of samples, we used eleven universal primer pairs to amplify eleven adjacent and overlapping fragments (*Simon et al., 2006*; *Zhang et al., 2008*; *Zhang et al., 2018a*; *Zhang et al., 2018b*). Then specific primers were designed from the initial overlapping fragments using Primer Premier 5.0 (Premier Biosoft International, Palo Alto, CA). A total of 45 pairs of primers were used in the present study to amplify and sequence the remaining gaps (Table S1). The cycling conditions and reaction volume of PCR amplifications were as in *Cheng et al. (2016)* and *Gao et al. (2018)*. All PCR products were sequenced by Sangon Biotech Company (Shanghai, China).

### Mitogenome annotation and sequence analyses

Manual proofreading and assembling of contiguous and overlapping sequences used DNASTAR Package v.6.0 (*Burland, 2000*). We annotated the tRNA genes by MITOS (freely available at http://mitos.bioinf.uni-leipzig.de/index.py) (*Bernt et al., 2013*). Two rRNA genes and the A + T-rich region were identified using the Clustal W in Mega 7.0 (*Kumar, Stecher & Tamura, 2016*) based on alignments of homologous sequences from other species of Cerambycidae available in GenBank (*Kim et al., 2009*; *Fang et al., 2016*; *Lim et al., 2017*). The nucleotide sequences of the 13 protein-coding genes (PCGs) were translated into amino acids based on the invertebrate mitogenome genetic code (*Cameron, 2014*). We used Mega 7.0 (*Kumar, Stecher & Tamura, 2016*) to find the open reading frames of the 13 PCGs and calculate AT content along with codon usage for the five newly sequenced mitogenomes. Circular mitogenome maps were generated by CG View server V 1.0 (*Grant & Stothard, 2008*). Composition skew analysis was calculated on the basis of the formula AT-skew = (A − T)/(A + T) and GC-skew = (G − C)/(G + C) (*Perna & Kocher, 1995*). Tandem Repeat Finder V 4.07 (http://tandem.bu.edu/trf/trf.html) (*Benson, 1999*) was used to find tandem repetitive sequences.

### Phylogenetic analyses

For the purpose of reconstructing the phylogenetic relationships of Cerambycidae, a nucleotide dataset (13P26) of the 13 protein-coding genes of 26 complete mitogenomes
was used (Table 1) according to the methods of *Zhang et al. (2019)*, this included the 5 newly determined sequences and 18 published complete mitogenomes of Cerambycidae (*Kim et al., 2009*; *Chiu et al., 2016*; *Fang et al., 2016*; *Guo et al., 2016*; *Li et al., 2016a*; *Li et al., 2016b*; *Wang et al., 2016*; *Lim et al., 2017*; *Liu et al., 2017*; *Song et al., 2017*; *Liu et al., 2018*; *Que et al., 2019*; *Wang et al., 2019*). Three species of Galerucinae, *Paleosepharia posticata*, *Diabrotica barberi* and *Diabrotica virgifera* served as the outgroups (*Coates, 2014*; *Wang & Tang, 2017*). To verify whether the lack of samples affects the relationships among the Cerambycidae, we reconstructed Cerambycidae phylogeny based on the nucleotide data (12P38) of 12 PCGs (omitting the *nad2* gene) from 38 complete or nearly complete mitogenomes (Table 1). These include all species of the 13P26 dataset, 8 directly submitted partial mitogenomes of Cerambycidae, one mitogenome of Necydalinae, two mitogenomes of Vesperidae and one mitogenome of Disteniidae (*Nie et al., 2017*). Each of the 13 protein-coding genes in 13P26 dataset or 12 protein-coding genes in 12P38 dataset was aligned using Clustal W in the program Mega 7.0 (*Kumar, Stecher & Tamura, 2016*). Conserved regions were identified by the program Gblock 0.91b (*Castresana, 2000*). Protein-coding genes were partitioned a priori by codon position. Accodrding to the analyses methods of *Zhang et al. (2008)*, *Ma et al. (2015a)*, *Ma et al. (2015b)* and *Cheng et al. (2016)*, we excluded the third codon positions because of the saturated third codon positions and obtained a 12P38 dataset with 5584 nucleotide sites and 13P26 dataset with 6960 nucleotide sites. So 12P38 dataset with 24 partitions and 13P26 dataset with 26 partitions were used. The optimal partitioning scheme and best-fitting models were selected by the program PartitionFinder 1.1.1 (*Lanfear et al., 2012*) based on the Bayesian information criterion (BIC) (Tables 2 and 3). Bayesian Inference (BI) and Maximum likelihood (ML) methods were used for phylogenetic analyses. BI analyses were carried out in MrBayes 3.2 (*Ronquist et al., 2012*) with the model of GTR + I + G. The runs were set for 10 million generations with sampling every 1,000 generations. The first 25% of generations were removed as burn-in and the average standard deviation of split frequencies in Bayesian was below 0.01. ML analyses were performed by RAxML 8.2.0 with the best-fitting model of GTRGAMMAI. Branch support values were inferred from a rapid bootstrap method applied with 1,000 replications (*Stamatakis, 2014*).

# RESULTS AND DISCUSSION

## Mitogenome organization and composition

In this study, the complete mitogenomes of five species of the subfamilies Cerambycinae and Lamiinae (*O. yaoshana, T. croccocincta, B. succinctor, N. carinicollis, Pterolophia* sp. ZJY-2019) were determined. Structures of the five newly sequenced entire mitogenomes are shown in Figs. S1–S5. The lengths of the five mitogenomes were basically within the range of the published Cerambycidae species in the GenBank database, covering sizes between 15,503 bp in *T. croccocincta* to 16,063 bp in *Pterolophia* sp. ZJY-2019. Every mitogenome of the five species possessed similar compositional profiles and featured the typical gene arrangement and orientation that have been hypothesized for most coleopteran insects (*Wolstenholme, 1992*; *Boore, Lavrov & Brown, 1998*), with the *trnW*-*trnC*-*trnY* triplet

**Table 1  Species used to construct the phylogenetic relationships along with GenBank accession numbers.**

| Order | Family | Species | GenBank No. | References |
|---|---|---|---|---|
| Cerambycidae | Lamiinae | *Anoplophora glabripennis* | DQ768215 | *Fang et al. (2016)* |
| | | *Psacothea hilaris* | FJ424074 | *Kim et al. (2009)* |
| | | *Thyestilla gebleri* | KY292221 | *Yang et al. (2017)* |
| | | *Monochamus alternatus* | KJ809086 | *Li et al. (2016a)* |
| | | *Anoplophora chinensis* | KT726932 | *Li et al. (2016b)* |
| | | *Apriona swainsoni* | NC_033872 | *Que et al. (2019)* |
| | | *Batocera lineolata* | MF521888 | *Liu et al. (2017)* |
| | | *Oberea yaoshana* | MK863509 | This study |
| | | *Thermistis croccocincta* | MK863511 | This study |
| | | *Blepephaeus succinctor* | MK863507 | This study |
| | | *Pterolophia* sp.ZJY-2019 | MK863510 | This study |
| | | *Olenecamptus subobliteratus*[*] | KY796054 | Directly submitted |
| | | *Eutetrapha metallescens*[*] | KY796053 | Directly submitted |
| | Cerambycinae | *Xylotrechus grayii* | NC_030782 | *Guo et al. (2016)* |
| | | *Xystrocera globosa* | MK570750 | *Wang et al. (2019)* |
| | | *Nortia carinicollis* | MK863508 | This study |
| | | *Massicus raddei* | KC751569 | *Wang et al. (2016)* |
| | | *Aeolesthes oenochrous* | AB703463 | *Chiu et al. (2016)* |
| | | *Obrium* sp. NS-2015 | KT945156 | *Song et al. (2017)* |
| | | *Pyrrhidium sanguineum*[*] | KX087339 | Directly submitted |
| | | *Chlorophorus simillimus*[*] | KY796055 | Directly submitted |
| | Prioninae | *Callipogon relictus* | MF521835 | *Lim et al. (2017)* |
| | | *Dorysthenes paradoxus* | MG460483 | *Liu et al. (2018)* |
| | | *Aegosoma sinicum* | NC_038089 | Directly submitted |
| | Lepturinae | *Leptura arcuata*[*] | KY796051 | Directly submitted |
| | | *Stictoleptura succedanea*[*] | KY796052 | Directly submitted |
| | | *Rhagium mordax*[*] | JX412743 | Directly submitted |
| | | *Stenurella nigra*[*] | KX087348 | Directly submitted |
| | | *Cortodera humeralis* | KX087264 | Directly submitted |
| | | *Anastrangalia sequensi* | KY773687 | Directly submitted |
| | | *Brachyta interrogationis* | KX087246 | Directly submitted |
| | Necydalinae | *Necydalis ulmi*[*] | JX220989 | Directly submitted |
| Disteniidae | Disteniinae | *Disteniinae* sp. BMNH 899837 | KX035158 | Directly submitted |
| Vesperidae | Philinae | *Spiniphilus spinicornis* | KT781589 | *Nie et al. (2017)* |
| | Vesperinae | *Vesperus conicicollis*[*] | JX220996 | Directly submitted |
| Chrysomelidae | Galerucinae | *Paleosepharia posticata* | KY195975 | *Wang & Tang (2017)* |
| | | *Diabrotica barberi* | KF669870 | *Coates (2014)* |
| | | *Diabrotica virgifera* | KF658070 | *Coates (2014)* |

**Notes.**
  [*]Partial genome.

**Table 2  The partition schemes and best-fitting models selected of 13 protein-coding genes in 13P26 data.**

| | Nucleotide sequence alignments | |
|---|---|---|
| Subset | Subset partitions | Best model |
| Partition 1 | atp6_pos1, cox1 pos 1, cox2_pos1, cox3_pos1, cytb_pos1 | GTR + I + G |
| Partition 2 | atp6_pos2, cox1_pos2, cox2_pos2, cox3_pos2, cytb_pos2, nd3_pos2 | TVM + I + G |
| Partition 3 | atp8_pos1, atp8_pos2, nd2_pos2, nd3_pos3, nd6_pos2 | GTR + I + G |
| Partition 4 | nd1_pos1, nd4l_pos1, nd4_pos1, nd5 pos1 | GTR + I + G |
| Partition 5 | nd1_pos2, nd4_pos2, nd4l_pos2, nd5_pos2 | GTR + I + G |
| Partition 6 | nd2_pos2, nd3_pos2, nd6_pos2 | TVM + I + G |

**Table 3  The partition schemes and best-fitting models selected of 12 protein-coding genes in 12P38 data.**

| | Nucleotide sequence alignments | |
|---|---|---|
| Subset | Subset partitions | Best model |
| Partition 1 | atp6_pos1, cox2_pos1, cox3_pos1, cytb_pos1 | GTR + I + G |
| Partition 2 | atp6_pos2, cox2_pos2, cox3_pos2, cytb_pos2, nd3_pos2 | TVM + I + G |
| Partition 3 | atp8_pos1, atp8_pos2, nd6_pos2 | HKY + G |
| Partition 4 | cox1 pos 1 | SYM + G |
| Partition 5 | cox1_pos2 | F81 + G |
| Partition 6 | nd1_pos1, nd4l_pos1, nd4_pos1, nd5 pos1 | GTR + I + G |
| Partition 7 | nd1_pos2, nd4_pos2, nd4l_pos2, nd5_pos2, | GTR + I + G |
| Partition 8 | nd3_pos1, nd6_pos1 | GTR + I + G |

(Tables S2–S6). Twenty-three genes were coded on the majority strand (J-strand), with the remaining fourteen genes coded on the minority strand (N-strand) (Figs. S1–S5). The nucleotide composition of the five longicorn beetle mitogenomes was strongly biased towards A and T, which made up 73.2% (*N. carinicollis*) to 79.1% (*O. yaoshana*) of the base pairs. A comparison of AT-skew and GC-skew showed that the AT skew of all mitogenomes was positive and the GC-skew was negative (Table 4).

## Protein-coding genes and codon usages

The orientations of the 13 the PCGs of the five longicorn beetles were identical to most coleopteran species (Tables S2–S6). Conventional initiation codons were assigned to the majority of the PCGs, except for *nad1*, which started with TTG in all five beetles. Most putative protein sequences showed typical stop codons (TAA/TAG), but the *nad4* and *nad5* genes of *O. yaoshana*, *T. croccocincta*, *B. succinctor* used a single T residue as the terminal codon. The *cox1* and *cox2* genes of *O. yaoshana*, *T. croccocincta* and *Pterolophia* sp. ZJY-2019 also used a single T residue as the terminal codon. Functional terminal codons can be produced by partial terminal codons in polycistronic transcription cleavage and polyadenylation processes (*Anderson et al., 1981*; *Ojala, Montoya & Attardi, 1981*; *Du et al., 2016*). The relative synonymous codon usage (RSCU) of the five Cerambycidae

Wang et al. (2019), *PeerJ*, DOI 10.7717/peerj.7633

**Table 4  Base composition of Cerambycidae mitochondrial genomes.**

| Species | A + T(%) | | | | AT-skew | | | | GC-skew | | | |
|---|---|---|---|---|---|---|---|---|---|---|---|---|
| | **Mito** | **PCGs** | **rRNAs** | **AT-richregion** | **Mito** | **PCGs** | **rRNAs** | **AT-richregion** | **Mito** | **PCGs** | **rRNAs** | **AT-richregion** |
| *O. yaoshana* | 79.1 | 77.8 | 81.1 | 87.1 | 0.03 | 0.14 | 0.04 | 0.04 | 0.20 | 0.01 | 0.38 | 0.24 |
| *T. croccocincta* | 76.4 | 76.4 | 78.6 | 87.4 | 0.15 | 0.15 | 0.04 | 0.04 | 0.13 | 0.01 | 0.49 | 0.45 |
| *B. succinctor* | 75.3 | 73.2 | 78.6 | 86.2 | 0.023 | 0.17 | 0.06 | 0.02 | 0.26 | 0.02 | 0.39 | 0.32 |
| *N. carinicollis* | 73.2 | 71.1 | 75.7 | 80.3 | 0.10 | 0.17 | 0.16 | 0.07 | 0.18 | 0.03 | 0.36 | 0.21 |
| *Pterolophia* sp.ZJY-2019 | 76.7 | 75.1 | 81.7 | 82.8 | 0.02 | 0.18 | 0.02 | 0.04 | 0.22 | 0.04 | 0.36 | 0.18 |

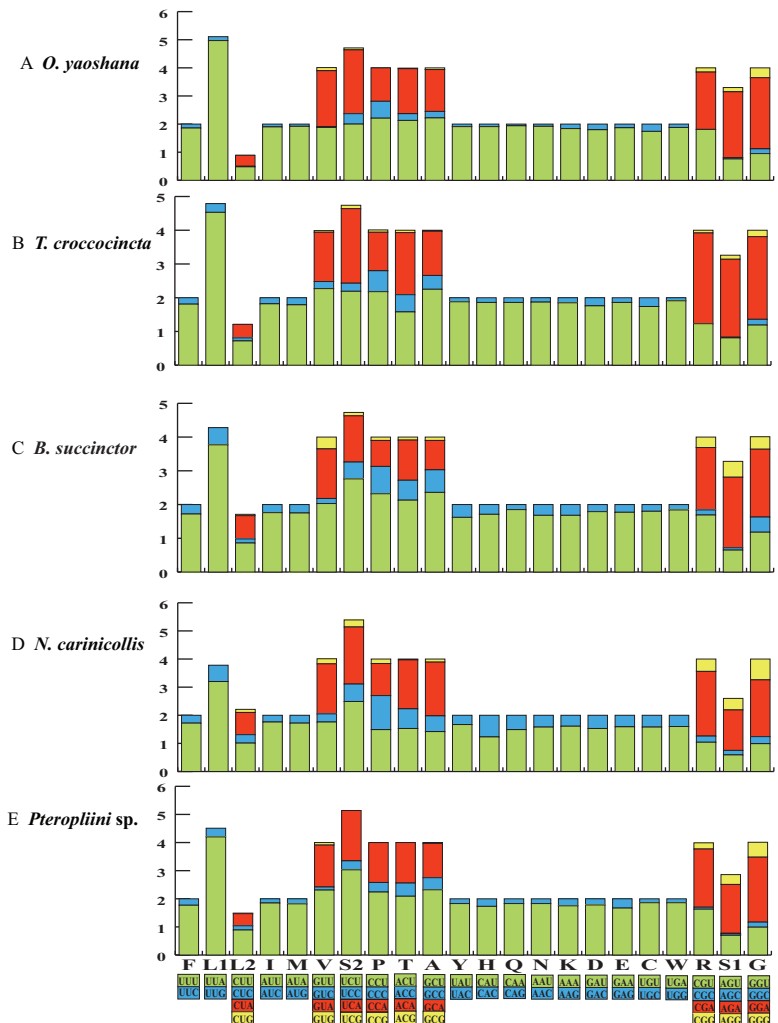

**Figure 1  The RSCU of five longicorn beetle mitochondrial genomes.** Codon families are provided on the *x*-axis along with the different combinations of synonymous codons that code for that amino acid. RSCU (relative synonymous codon usage) is defined on the *Y* axis.

mitochondrial genomes was calculated (Fig. 1, Table S7). The results showed an over-utilization of A or T nucleotides in the third codon position as compared to other synonymous codons, this is normally considered to be caused by genome bias, optimum choice of tRNA usage or the benefit of DNA repair (*Chai & Du, 2012*; *Ma et al., 2015a*; *Ma et al., 2015b*).

Comparative analyses also indicated that the major customarily utilized codons and the codon usage patterns of the five samples were conservative. For instance, each of the five mitogenomes possessed UUA (Leu), AUU (Ile), UUU (Phe), and AUA (Met) as the most frequently used codons. All codons contained A or T nucleotides, indicating that the strong AT mutation bias obviously influenced the codon usage (*Powell & Moriyama, 1997*; *Rao et al., 2011*). Furthermore, the codons rich in AT encoded the most abundant amino

acids, e.g., Leu (15.6–16.4%), indicating that the AT bias also influences the amino acid constituents of the proteins encoded by the mitochondrial genes (*Foster, Jermiin & Hickey, 1997*; *Min & Hickey, 2007*).

## Ribosomal RNAs and transfer RNAs

The two expected rRNAs (16S rRNA and 12S rRNA) were found in the mitochondrial genomes of all five longicorn beetles. The 16S rRNA gene was situated between *trnL* and *trnV* whereas the 12S rRNA gene was between *trnV* and the A + T-rich region. Due to the impossibility of faultless determination by DNA sequence alone, the terminus of the rRNA genes in coleopteran mitogenomes has been presumed to stretch to the border of the flanking genes (*Boore, 2001*). Therefore, the 16S rRNA was presumed to fill the blank between *trnL* and *trnV* whereas the border between 12S rRNA and the putative A + T-rich region was defined based on alignments of homologous sequences of known longicorn beetles (*Boore & Brown, 2000*). The sizes of 16S rRNA in the five beetle mitogenomes varied from 1261 bp for *N. carinicollis* to 1283 bp for *O. yaoshana*, and the sizes of 12S rRNA ranged between 759 bp for *Pterolophia* sp. ZJY-2019 to 787 bp for *T. croccocincta*. These fit within the lengths detected in other coleopteran mitogenomes. The A + T content of the rRNA genes was the highest (81.7%) in the *Pterolophia* sp. ZJY-2019 mitogenome and the lowest in the *N. carinicollis* mitogenome (75.7%). The AT-skew of 16S rRNA and 12S rRNA showed great positivity, whereas the GC-skew was somewhat negative (Table 4), which indicated the occurrence of less As and Cs than Ts and Gs (*Eyrewaker, 1997*).

The 22 typical tRNAs were detected in all five species like other published longicorn beetles. All the anticodons were also highly conserved compared to other beetle species. Twenty-two tRNAs excluding *trnS1* displayed the classic clover-leaf secondary structure, whereas *trnS1* lacked the dihydrouridine (DHU) arm and formed a simple loop (Fig. S6). Nevertheless, this abnormal tRNA has proven to be functional, although somewhat less effective than conventional tRNAs (*Steinberg & Cedergren, 1994*; *Hanada et al., 2001*; *Stewart & Beckenbach, 2003*). Another unusual feature was the use of TCT as the *trnS1* anticodon in Cerambycidae, whereas most arthropods use a GCT anticodon in *trnS1*. In many other coleopteran mitogenomes the *trnS1* anticodon (TCT) can also be observed (*Friedrich & Muqim, 2003*; *Bae et al., 2004*). Mismatched pairs also exist in stems of tRNAs. For example, the mismatched pairs U-G existed in the DHU stem of *trnY* and *trnQ*; U-U existed in the T ΨC stem of *trnC* and in the anticodon stem of *trnL1*; G-U existed in acceptor stem of *trnC*. It has been verified that mismatched pairs can be revised via editing processes or may symbolize abnormal pairings (*Negrisolo, Babbucci & Patarnello, 2011*).

## A + T-rich region

A large non-coding region between 12S rRNA and *trnI*, ranging between 861 bp for *O. yaoshana* to 1137 bp for *Pterolophia* sp. ZJY-2019, was found in the mitogenomes of the five beetles. Owing to the high AT content levels of the overall mitogenome, this non-coding element was defined as the A + T-rich region. It has been verified that the A + T-rich region harbors the origin sites and essential regulatory elements for transcription and

replication (*Wolstenholme, 1992*; *Taanman, 1999*; *Yukuhiro et al., 2002*; *Saito, Tamura & Aotsuka, 2005*). The sequence of this region is relatively conserved owing to its high A + T content, and thus it is impossible to use as a molecular marker (*Zhang & Hewitt, 1997*). The existence of tandem repeats in the mitochondrial A + T-rich region has been observed in many coleopteran species. Some studies such as that conducted by *Sheffield et al. (2008)* have shown that the A + T-rich region of *Trachypachus holmbergi* (Coleoptera: Trachypachidae) possessed 21 similar copies of tandem repeats consisting of a 58-bp fragment. The A + T-rich region of *Priasilpha obscura* (Coleoptera: Phloeostichidae) is known to possess 6 tandem repeats of a 132-bp fragment and *Psacothea hilaris* (Coleoptera: Cerambycidae) possesses 7 identical copies of a 57 bp tandem repeat (*Kim et al., 2009*). In the present study, we found tandem repetitive sequences in all five newly sequenced mitogenomes. The mitogenomes of *T. croccocincta* and *B. succinctor* contained three copies of tandem repetitive sequences with lengths of 19 and 43 bp, respectively. Four tandem repeats of a 19-bp fragment were found in the mitogenome of *Pterolophia* sp. ZJY-2019, whereas two tandem repeats of a 25-bp fragment existed in *N. carinicollis*. The tandem repeats generally exhibited high A + T contents. Moreover, two poly-T stretches were detected in the mitogenome of *N. carinicollis*: one stretch was 16 bp in length (position: 14,880–14,895) near the 12S rRNA gene and the other stretch was 17-bp in length (position: 15,283–15,299). Previous studies have confirmed that the two poly-T stretches were structural signals for the recognition of proteins that performed a role in replication initiation (*Andrews, Kubacka & Chinnery, 1999*).

## Intergenic regions

The mitogenomes of *O. yaoshana*, *T. croccocincta*, and *N. carinicollis* contain 6, 7, 9 non-coding intergenic spacer sequences, with total lengths of 28 bp, 28 bp, and 31 bp, respectively, whereas *B. succinctor* has 8 non-coding intergenic spacer sequences of 52 bp in total length. Unexpectedly, a total of 354-bp of intergenic spacer, whose elements ranged from 1 to 184 bp in length was found in the mitogenome of *Pterolophia* sp. ZJY-2019. The sequences are divided into 9 regions, containing two large intergenic spacers. The largest one is 184 bp long situating between *trnC* and *trnY*, and the other is 157 bp long situated between *trnS2* and *nad1* (Table S6). Consequently, the total length of the mitogenome of *Pterolophia* sp. ZJY-2019 is longer than that of other longicorn beetle species. The longer mitogenome length is due to the existence of its extended large intergenic spacers not the A + T-rich region. Previously reported tandem repeat units or additional origins of replication have been identified within this region (*Dotson & Beard, 2001*; *Rodovalho et al., 2014*). Proven by the lack of introns, rare intergenic spacers, defective terminal codons and overlapping fragments, mitogenomes characteristically show exceptional compactness of organization (*Ojala, Montoya & Attardi, 1981*). Nevertheless, according to *Yuan et al. (2016)* and *Haddad et al. (2018)*, large non-coding regions (except the A + T-rich region) in mitochondrial genomes were observed in *Pyrocoelia rufa* (Coleoptera: Lampyridae) and some *Hycleus* species (Coleoptera: Meloidae). Coincidentally, a 5 bp consensus motif (TACTA) exists in the intergenic regions situated between *trnS2* and *nad1* of all five species studied here. This pentanucleotide motif is conserved across coleopteran

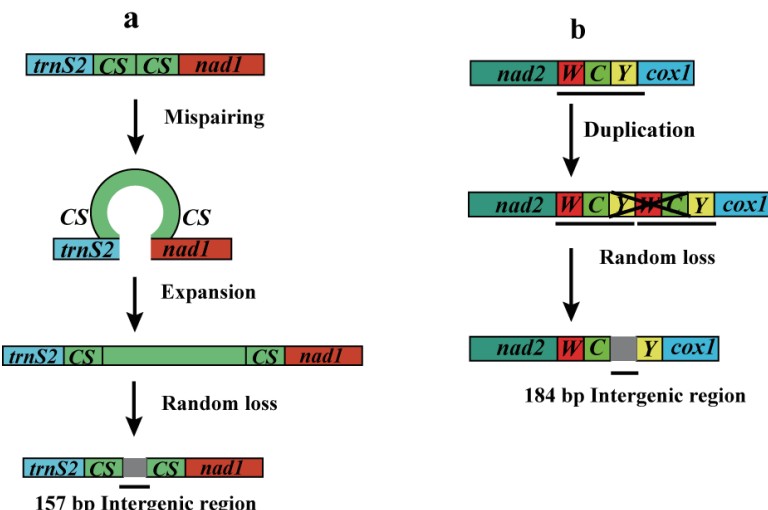

**Figure 2** **Putative mechanisms for formation of the two large intergenic regions (IGRs) that exist in** *Pterolophia* **sp. ZJY-2019.** (A) The slipped-strand mispairing and random loss model to explain the 157 bp-IGR between *trnS2* and *nad1*. The CS indicates the 18 bp conservative sequence TTACTAAATTTAAT-TAACTAAA. (B) The duplication/random loss model to explain the 184 bp-IGR between *trnC* and *trnY*.

lineages (*Kim et al., 2009*; *Liu et al., 2018*), similar to the findings that *Evania appendigaster* (Hymenoptera: Evaniidae) possessed a 6 bp motif 'THACWW' and *Chilo suppressalis* (Lepidoptera: Pyralidae) possessed a 7 bp motif 'ATACTAA', respectively (*Wei et al., 2010*; *Gong et al., 2018*).

In the mitogenome of *Pterolophia* sp. ZJY-2019, the large intergenic region was situated between *trnS2* and *nad1*, which included two copies of a 22 bp long consensus sequence (TTACTAAATTTAATTAACTAAA) in both ends of the intergenic region. The formation of an intergenic region may be explained by slipped-strand mispairing (*Levinson & Gutman, 1987*; *Du et al., 2017*). Based on this theory, mispairing occurred during replication of DNA strands, and what followed next was misaligned reassociation and then replication or repair was caused by insertions of several repeat units. The resulting tandem repeat underwent random loss and/or point mutation, with only the repeat units in both extremities remaining (Fig. 2A). However, a tandem repeat was not found in the intergenic region located between *trnC* and *trnY* of *Pterolophia* sp. ZJY-2019. We conjectured that some errors in DNA replication can lead to tandem duplication in tRNA clusters *of trnW-trnC-trnY*, followed by the random loss of partial duplicated genes, and leading to the large intergenic region formed by the residues (Fig. 2B). In addition, *Hua et al. (2008)* suggested that the duplication-random loss model caused the rearrangements in Hemiptera. *Du et al. (2017)* also suggested that the duplication-random loss model was an evolutionary ancient mechanism in Coleoptera, which led to the random loss of nucleotides.

Consequently, compared to the original tRNAs, the residual intergenic region was not conserved. According to *Du et al. (2017)*, four species of *Hycleus* genera harbored similar location and sequence of non-coding regions, which indicated that the region may serve as
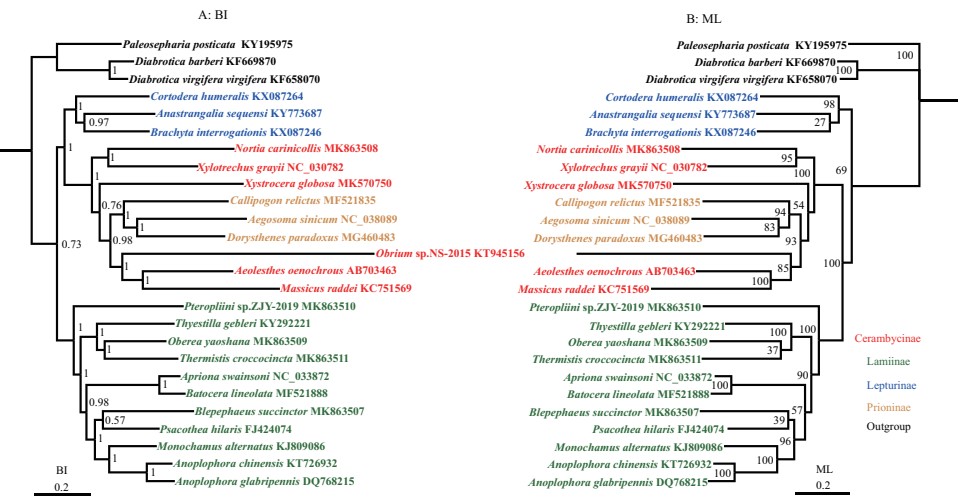

**Figure 3** **Phylogenetic relationships of Cerambycidae in BI and ML analyses.** The data includes 23 species of Cerambycidae as the ingroup and three species of Chrysomelidae as the outgroup. The GenBank accession numbers of all species are also shown.

a latent symbol to distinguish *Hycleus* from the other genera. Thus, we speculated the large intergenic region of *Pterolophia* sp. ZJY-2019 may be a molecular feature in *Pterolophia*, though we were unable to adequately confirm it owing to the lack of enough samples.

## Phylogenetic analyses

The phylogenetic relationships were reconstructed based on the nucleotide data (13P26) with BI and ML methods (Fig. 3). BI and ML phylogenetic analyses yielded a similar topology except for the position of Lepturinae, which was in the sister group of (Cerambycinae + Prioninae) with high values in BI, but supported as the basal group of Cerambycidae in ML analyses. The BI tree indicated that Cerambycidae split into 2 major groups (0.73): a clade of (Lepturinae + (Cerambycinae + Prioninae)) and a clade of Lamiinae. The monophyly of Lamiinae, Lepturinae and Prioninae was supported by both BI and ML analyses, whereas the monophyly of Cerambycinae was not recovered. Within the subfamily Lamiinae, the clade of (Lamiinae + (*Batocera lineolata* + *Thyestilla gebleri*)) was supported. However, *Liu et al. (2018)* favoured *T. gebleri* as the basal position of Lamiinae with a high value, and *B. lineolata* and *Apriona swainsoni* were reliably recovered as a sister group. Our results concurred with the suggestion that *B. lineolata* was closely related to *A. swainsoni*, rather than *T. gebleri*. The results also placed *Pterolophia* sp. ZJY-2019 as a sister group of all remaining Lamiinae. Moreover, our results suggested that *O. yaoshana* clustered with *Trachypachus holmbergi*, as a sister group of *T. gebleri*. For the relationship within Cerambycinae, *M. raddei*, *A. oenochrous* and *Obrium* sp. NS-2015 were gathered into one clade and most closely related to the subfamily Prioninae rather than the remaining Cerambycinae, consistent with the morphological and molecular analyses in previous reports (*Liu et al., 2018*).

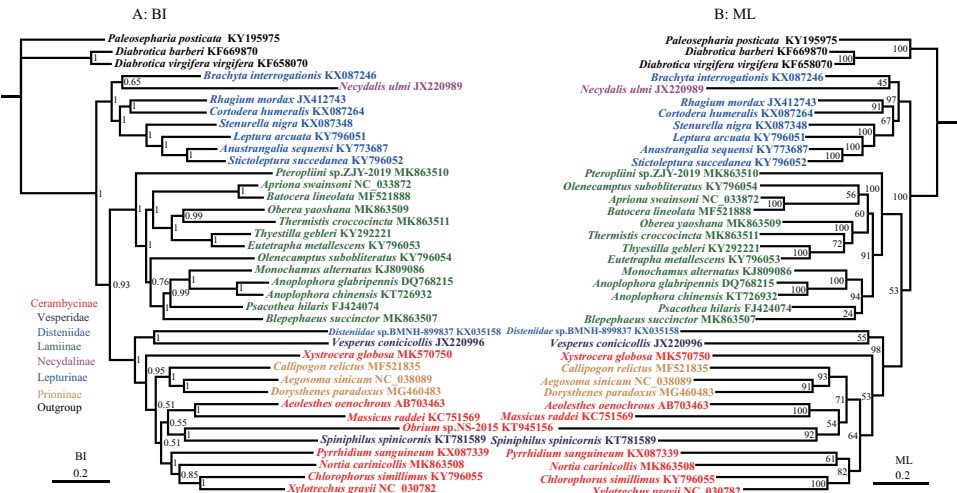

**Figure 4** **Phylogenetic relationships of Cerambycidae in BI and ML analyses.** The data includes 35 species of Cerambycidae as the ingroup and three species of Chrysomelidae as the outgroup. The GenBank accession numbers of all species are also shown.

The results from the BI trees of the nucleotide dataset showed that Lepturinae cluster with the clade (Cerambycinae + Prioninae) with a high support value (Fig. 3). However, in the ML tree, a close relationship between Lamiinae and (Cerambycinae + Prioninae) was supported with 100% posterior probabilities (Fig. 3). The relationship between Cerambycinae and Prioninae is not currently understood in great detail. Prioninae were traditionally considered basal in Cerambycidae by morphology (*Hatch, 1958*; *Svacha, Wang & Chen, 1997*; *Farrell, 1998*). In addition, *Hunt et al. (2007)* and *Haddad et al. (2018)* pointed out that Prioninae could be placed at the basal position of Cerambycidae based on molecular phylogenetic studies. However, in BI and ML analyses of the 13P26 dataset Prioninae clustered into Cerambycinae, which was consistent with the phylogenetic position of Prioninae recovered by *Raje, Ferris & Holland (2016)*.

The most controversial point in our results was in Cerambycinae (Fig. 3), which was represented by five different genera and rendered non-monphyletic in Prioninae. However, Cerambycinae was not supported as monophyletic based on molecular by *Liu et al. (2018)* and *Haddad et al. (2018)*, but was recovered in other molecular studies (*Lim et al., 2017*; *Liu et al., 2017*).

To further discuss the monophyly of subfamilies within Cerambycidae, more samples were needed to confirm and rebuild the phylogenetic relationship of Cerambycidae using 12 protein-coding genes. The phylogenetic relationships were reconstructed based on the nucleotide data (12P38) with BI and ML methods (Fig. 4). Prioninae still clustered into Cerambycinae in BI and ML analyses of the 12P38 dataset, which agreed with the phylogenetic position of Prioninae recovered using the 13P26 dataset. In BI and ML analyses, all trees recovered the monophyly of Lamiinae (although the relationships within Lamiinae were different). The Lamiinae formed a sister group to a clade comprising Disteniidae, Prioninae, Cerambycinae and Vesperidae. The clade of

Lepturinae and Necydalinae was a sister to the remaining species of Cerambycidae *s. l.* In addition, BI and ML analyses recovered the monophyly of Prioninae including *Callipogon relictus*, *Dorysthenes paradoxus* and *Aegosoma sinicum*, as proposed by *Wang et al. (2019)*. However, BI and ML results did not support the monophyly of Cerambycinae with respect to Prioninae and *Spiniphilus spinicornis* (Vesperidae). It has been well accepted that Necydalinae and Lepturinae have a close relationship. The monophyly of Lepturinae was recovered in both BI and ML analyses of the 13P26 dataset. However, BI and ML trees from the 12P38 dataset returned a paraphyletic Lepturinae, due to a sister relationship between *Necydalis ulmi* (Necydalinae) and *Brachyta interrogationis* (Fig. 4).

Previous studies recognized *S. spinicornis* as a species of Vesperinae in Cerambycidae (*Napp, 1994*). Nevertheless, subsequent studies considered it to belong to the subfamily Philinae of Vesperidae (*Svacha, Wang & Chen, 1997*; *Lin & Bi, 2011*; *Nie et al., 2017*). Further phylogenetic studies put *S. spinicornis* in the fairly controversial placements (*Bi & Lin, 2015*; *Liu et al., 2018*). In addition to our results, a recent molecular study also indicated a similar relationship (*Liu et al., 2018*).

## CONCLUSION

In this study, we present five completely sequenced mitogenomes of Cerambycidae. The five longicorn beetle species shared similar gene organization with the insects previously reported. The gene sequences and composition of the mitogenomes were relatively conservative with no rearrangements, duplications or deletions. Two large intergenic spacers existed in *Pterolophia* sp. ZJY-2019. The duplication/random loss model and slipped-strand mispairing may explain the existence of these regions. The phylogenetic results inferred from mitogenomes supported the monophyly of Lamiinae and Prioninae in BI and ML analyses, whereas the Cerambycinae and Lepturinae were recovered as non-monophyletic. Although data collected thus far could not resolve the phylogenetic relationships within Cerambycidae, this study will increase the richness of the Cerambycidae genome information and assist in phylogenetic, molecular systematics and evolutionary studies of Cerambycidae.

## ACKNOWLEDGEMENTS

We are grateful to Wen-Yong Feng for his help in sample collection.

### Funding

This research was supported by the Zhejiang provincial Natural Science Foundation (Y18C040006), the National Natural Science Foundation of China (31370042), the College students' Innovation and Entrepreneurship Project in China (No. 201810345043), the College students in Zhejiang Normal University Innovation and Entrepreneurship Plan (2018-317) for the study design, data collection and analyses. The funders had no role

in study design, data collection and analysis, decision to publish, or preparation of the manuscript.

### Grant Disclosures

The following grant information was disclosed by the authors:

Zhejiang provincial Natural Science Foundation: Y18C040006.

National Natural Science Foundation of China: 31370042.

Innovation and Entrepreneurship Project in China: 201810345043.

College students in Zhejiang Normal University Innovation and Entrepreneurship Plan: 2018-317.

### Competing Interests

Kenneth B. Storey and Jia-Yong Zhang are Academic Editors for PeerJ.

### Author Contributions

- Jun Wang and Xin-Yi Dai conceived and designed the experiments, performed the experiments, analyzed the data, contributed reagents/materials/analysis tools, prepared figures and/or tables, authored or reviewed drafts of the paper.
- Xiao-Dong Xu and Zi-Yi Zhang analyzed the data, prepared figures and/or tables, authored or reviewed drafts of the paper.
- Dan-Na Yu conceived and designed the experiments, analyzed the data, contributed reagents/materials/analysis tools, authored or reviewed drafts of the paper.
- Kenneth B. Storey authored or reviewed drafts of the paper.
- Jia-Yong Zhang conceived and designed the experiments, analyzed the data, contributed reagents/materials/analysis tools, authored or reviewed drafts of the paper, approved the final draft.

### Data Availability

Five new sequenced mitochondrial genomes are available at GenBank: MK863507–MK863511.

### Supplemental Information

Supplemental information for this article can be found online at http://dx.doi.org/10.7717/peerj.7633#supplemental-information.

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
