# Peer review of "The complete mitochondrial genomes of five longicorn beetles (Coleoptera: Cerambycidae) and phylogenetic relationships within Cerambycidae"

_PeerJ, doi:10.7717/peerj.7633_

## Round 0.1 · original submission · Minor Revisions

In my opinion this manuscript needs only minor revisions. For instance, some scientific terms need to be revised (e.g. paraphyly vs non-monophyly), some or clearer info should be provided (e.g. the partition schemes and models inferred by PartitionFinder; pre-settings for partitions of alignments; if additional primers were used as sequencing primers over the 11 amplicons or not). Authors should also clarify why they choose to use only nucleotide data and not aminoacid sequences.
I also believe that the findings about the mitogenome of Pterolophia sp (i.e., is the presence of a large intergenic region) is particularly interesting but unfortunately the authors cannot confirm if this represents a synapomorphy.

·

Basic reporting

Dr Wang and collaborators have sequenced five new mitogenomes of cerambycids. The manuscript presents a description of the genomes and a phylogenetic reconstruction of the group. The genomes are well described and provide some insight into the genome evolution in the group. The phylogenetic reconstructions provide some new insights, even if they are not conclusive (something that is nevertheless not unexpected given the complexity of the group).
The genomic section is well structured and clear. The phylogenetic analyses are standard in the field. The introduction and discussion, with ample reference to preexisting literature, are above standard for this type of studies.

I encourage the Editor to seek for advice from a specialist of cerambycids, as I am not in the position to evaluate this part of the study.
In the end, I would suggest publication after minimal editing (see comments above).

- Language is clear and correct.
- Intro and backgroud are clear and reference to preexisting literature is ample.
- Structure of the manuscript seems to conform to PeerJ standards.
- Figures are relevant and high quality (minor comments below).
- Sqn files covering all new genomes with annotations are proveded as raw data.

Experimental design

- Research is within the scopes of the journal
- Research question well defined and relevant for the currect knowledge of cerambycid mitochondrial genomics.
- Research is conducted according to ‘standard’ methodological standards in the field.
- Methods are described in sufficient detail (minor comments below)

Validity of the findings

- Impact and novelty not assessed, as from the journal’s instruction to reviewers.
- Sequence data have been provided and, most important, will be made available through GenBank. - Correctness of sequence data is assumed. Given the twist of OpenJ towards ‘openness’, I would encourage the authors to make the two alignments available through the journal’s website.
- Speculation is limited and identifiable.
- Conclusions are well in line with the results.

Additional comments

Lines 93-94: please rewrite.
Lines 112-114: it is not clear wether the additional primers were used as sequencing primers over the 11 amplicons or to amplify/sequence additional fragments.
Line 151: how were starting minimal partitions for PartitionFinder identified?
Lines 154-156: please confirm that these models are the ones identified by PartitionFinder.
Line 157: 10 million generations may not be appropriate for a dataset of this size. Please check traces in Tracer to confirm optimal convergence.
Line 170 and elsewhere: please refer to the ancestral Pancrustacean arrangement as such (to benefit the non specialist reader).
Line 239: please define ‘conserved’. In structure? In sequence? In molecular features? The notion of the AT rich region is conserved sounds a little strange.
Line 251: why ‘unexpectedly’?
Line 264: randomly meaning equally?
Line 294: did the authors observe any similarity in sequence between the spacer and any of the neighbouring tRNAs that would support their model?
Tables S1-7: please check column formatting for uniformity (this may be a file conversion problem)

Figure 1: column height (sum of two columns for S and L if collectively analyzed) should, in my understanding, sum to the number of different codons for a given aminoacid. S in the last panel, on the other hand, seems to sum to 7.5 (in figure as in table). Please check.
Figure S6. Naming tRNAs using alphabetic letters is useless and slightly confusing. I would valorize the grid pattern of this figure naming columns (species) and rows (tRNAs) instead.
Figures S1-S5. Considering that base composition/skews are not very informative at the nucleotide scale, and that gene order is conserved, the authors may evaluate the possibility to simplify figures into one single collective schematic representation of genomes.
Figure 2: the second step of panel a is not totally clear. Didn’t the authors hypothesize an incorrect match between the two copies of the repeat?
Figures 3-4: figures appear at slightly low resolution in my pdf. Would it be possible to have a better image definition is the final output?

Reviewer 2 ·

Basic reporting

no comment

Experimental design

Wang et al. present a good paper looking at mitochondrial phylogenomics of Cerambycidae, adding additional five longicorn beetle mitogenomes. This work contributes to the reconstruction of higher-level phylogeny of Cerambycidae. The large intergenic region identified in the new mitogenomes is interesting. Whether this feature can serve as a synapomorphy of Pterolophia need further sequencing more mitogenomes in this group. Whether this feature can serve as a synapomorphy for the genus Pterolophia need sequencing more mitogenomes in this group.

Validity of the findings

no comment

Additional comments

Although this MS is pretty well written, some sentences need to be improved. I've made some editorial suggestions on a copy of the manuscript which will be attached to this review. This shouldn't be too hard to turn around for publication.

I have a couple of points which need addressing.
1) The Cerambycinae and Lepturinae was recovered to be non-monophyletic, because the Lepturinae is paraphyletic in Figure 4 and the Cerambycinae is polyphyletic in Figure 3-4. Thus, using non-monophyletic may be suitable to describe the Cerambycinae and Lepturinae.
2) The pre-settings for partitions of alignments should be provided. For example, you can set 15 blocks according to gene types or 41 blocks according to codon positions and gene types. Additionally, the partition schemes inferred by PartitionFinder should also be presented in this paper.

Annotated reviews are not available for download in order to protect the identity of reviewers who chose to remain anonymous.

Reviewer 3 ·

Basic reporting

Clear and unambiguous, professional English used throughout

Experimental design

Methods described with sufficient detail & information to replicate

Validity of the findings

Conclusions are well stated, linked to original research question & limited to supporting results.

Additional comments

The manuscript entitled “The complete mitochondrial genomes of five longicorn beetles (Coleoptera: Cerambycidae) and phylogenetic relationships within Cerambycidae” was well organized and clearly written in terms of grammar and style. The most important scientific contribution of this study is five new mitochondrial genomes of Cerambycid species have been determined. This study attempts to reconstruct the mitochondrial phylogeny of Cerambycidae but has some issues unsolved. My suggestions are as follows:

1. In methods section, authors stated “The optimal partitioning scheme and best models were selected by the program PartionFinder 1.1.1…”, but the result of partitioning scheme and models have not been showed.

2. in RESULTS AND DISCUSSION section, the authors discussed the non-coding region and stated “According to Du et al. (2017), four species of Hycleus genera harbor similar lengths of large intergenic region…”, but this expression was not appropriated. Du et al. 2017 emphasized the location and sequence similarity of the non-coding region in Hycleus species, but not the lengths.

3. For phylogenetic analysis, why only nucleotide data were used? Why do not use amino acid sequences? Or both? Previous studies showed that the Coleoptera phylogeny is sensitive to different mitochondrial datasets and analytical methods. In my opinion, additional phylogenetic analyses with more datasets (amino acid and nucleotide with different partitions) and more analytical methods (ML, MrBayes and PhyloBayes) are necessary.

Some issues have not resolved in the phylogenetic analysis in this manuscript, and the authors have not attempted more. I think maybe some issues would be well recovered if the phylogenetic analyses were complemented with more datasets and more analytical methods (maybe not). But this would greatly improve the manuscript.

4. Several figures are of not good quality, and need to be improved.

---

## Round 0.2 · accepted · Accept

The current version of the manuscript has been significantly improved. The authors provided better explanations of methods and approaches used in the study, correct some unclear or not scientifically sound sentences and terms.

I think that the work can be now accepted for publication (in this version I noticed some typos throughout the text, but I guess that the authors still have the chance to correct them).